# Circumcising daughters in Nigeria: To what extent does education influence mothers' FGM/C continuation attitudes?

Josephine Akua Ackah[1,2]*, Patience Ansomah Ayerakwah[3], Kingsley Boakye[4], Bernard Afriyie Owusu[2], Vincent Bio Bediako[2], Millicent Gyesi[5], Edward Kwabena Ameyaw[6,7], Francis Appiah[4,8]

1 Department of Population Health, London School of Hygiene and Tropical Medicine, London, United Kingdom, 2 Department of Population and Health, University of Cape Coast, Cape Coast, Ghana, 3 Department of Optometry, University of Cape Coast, Cape Coast, Ghana, 4 School of Public Health, Kwame Nkrumah University of Science and Technology, Kumasi, Ghana, 5 Department of Science, St. Vincent College of Education, Yendi, Ghana, 6 Institute of Policy Studies and School of Graduate Studies, Lingnan University, Lingnan, Hong Kong, 7 L & E Research Consult Ltd, Wa, Upper West Region, Ghana, 8 Department of Social Sciences, Berekum College of Education, Berekum, Bono Region, Ghana

* Josephine.Ackah@lshtm.ac.uk

Data Availability Statement: We used data from the 2018 Nigeria Demographic and Health Survey. The data is freely available upon registration at

## Abstract

Education has been adjudged as an important behavioural change intervention and a key player in combating Female Genital Mutilation/Cutting (FGM/C). An assumed pathway is that it influences FGM/C attitudes. However, empirical evidence that explores this assumption is scarce. Hence, our study examines whether the associative effect of FGM/C continuation attitudes on circumcision of daughters is influenced by the level of a mother's education in Nigeria. We extracted data from the 2018 Nigeria Demographic and Health Survey (NDHS). The study focused on youngest daughters that were born in the last five years preceding the survey. A sample of 5,039 children with complete data on variables of interest to the study were analysed. The main outcome variable for this study is "circumcision among youngest daughters". The key explanatory variables were maternal "FGM/C continuation attitudes" and "education". At 95% confidence interval, we conducted a two-level logistic regression modelling and introduced interaction between the key independent variables. In the study's sample, the prevalence of FGM/C was 34%. It was lower for daughters whose mothers had higher education (12%) and believe FGM/C should discontinue (11.1%). Results from the multivariate analysis show statistically significant odds of circumcision for a daughter whose mother has had higher education and believes FGM/C should discontinue (OR-0.28, 95%CI: 0.08–0.98). For women who believe FGM/C should discontinue, the probability of daughter's circumcision reduced by 40% if the mother has attained higher education. Among those who believe FGM/C should continue, the probability of daughter's circumcision worsened if the mother had attained higher education (64%), however, this result was influenced by mothers' experience of circumcision. Education influences FGM/C attitudes, nonetheless, women's cutting experience can be a conduit for

https://dhsprogram.com/data/available-datasets.cfm or IPUMS-DHS (idhsdata.org).

**Funding:** The authors received no specific funding for this work.

**Competing interests:** The authors have declared that no competing interests exist.

which the practice persists. Promoting female education should be accompanied by strong political commitment towards enforcing laws on FGM/C practice.

## Introduction

Female Genital Mutilation/Cutting (FGM/C) is a "surgical procedure involving partial or total removal of the external female genitalia or other injury to the female genitalia organs, whether for cultural or any other non-therapeutic reasons" [1, 2]. FGM/C has received considerable attention in both local and international discussions due to its dehumanizing act and violation of women's health rights [3–7]. About 200 million women and girls have undergone FGM/C, and additional 68 million girls will undergo the cutting before 2030 if appropriate and timely interventions are not implemented [8]. FGM/C inflicts irreversible and devastating complications on women and they include sexual dysfunction, obstetric complications, psychological trauma [9–12], and death [13, 14]. Also, the use of unsterilized instruments in some contexts predisposes victims to human papillomavirus (HPV) and other infections [15]. As a result, FGM/C is globally recognized as a harmful cultural practice that needs to be eliminated [8].

In the face of increasing global attention, the tradition continues to be widespread and mostly prevails in developing regions with the highest prevalence of FGM/C in sub-Saharan Africa [7, 8] despite restrictions. FGM/C is deep-rooted in African culture and traditions largely due to social norms, quest to suppress female sexuality, aesthetic preferences, social cohesion and religion [16–18]. In Nigeria, the prevalence of FGM/C is high with about one in every five women (15–49) ever circumcised [8]. In the quest to achieve the Sustainable Development Goal (SDG) 5.3 aimed at eradicating all forms of harmful practices against women and children including FGM/C by the year 2030, the Nigerian government formulated and implemented laws, policies, programs and strategies to combat the practice. Notable are the Violence against Persons (Prohibition) Act 2015 (VAPP Act); the 2013/2017 National Policy and Plan for Action for Elimination of FGM/C in Nigeria as well as several anti-FGM/C campaign programs of which the UNFPA-UNICEF Joint Programme on Female Genital Mutilation [19] is noteworthy. "Odimma Nwanyi bu ka Chi Siri Ke" which translates as "wholeness of the female is as created by God" has been the Nigerian campaign slogan for the UNFPA-UNICEF joint programme [19].

The policies and programs have been beneficial, contributing to a decline in cases among women aged 15–49 years from 25 percent in 2013 to 20 percent in 2018 [20]. However, for girls aged 0–14 years, this has not been the case. Recent estimates reveal that the practice is on the rise from 16.9 percent in 2013 to 19.2 percent in 2018; a "worrying trend" [20]. Furthermore, majority of women in Nigeria are cut before reaching their fifth birthday [5]. Estimates from the 2018 Nigerian Demographic and Health Survey revealed that 86.5 percent of circumcised women aged 15–49 years were cut before reaching five years [21]. This implies that the first five years of life is a crucial period within which FGM/C occurs.

Several studies have investigated the underlying socio-demographic and economic predictors of the practice to inform policies and programs [1, 22–24]. Findings reveal that girls that belong to the Islamic religion; reside in poor households; live in rural areas; have mothers with no formal education and whose mothers have ever been circumcised are susceptible to the practice [1, 24]. On the basis of attitude, women who feel FGM/C should continue are more likely to circumcise their children [23] and in Nigeria, the proportion of such women (16-49years) is more than 20 percent [5, 21].

Maternal education plays an essential role in attitudinal and behavioural change, especially in eradicating FGM/C. Education exposes individuals to a wide information base that makes them more likely to adopt healthy behaviours [25]. Higher educational levels translate into better economic [26] and health [27] opportunities that impact women's health decisions. Women are therefore more likely to opt for the discontinuation of FGM/C if they are highly educated [28–30] with reduced chances of circumcision among their daughters [23]. The associative effect of education on FGM/C has, however, been contested. De Cao and Giulia [31] investigated the impact of the Universal Primary Education program in Nigeria on daughters' circumcision and found that despite the overall increase in years of education among women, there wasn't a corresponding statistically significant influence on attitudes towards FGM/C and daughters' circumcision. Segun [32] also shared that formal education and FGM/C are rather two parallel lines because a substantial number of educated women continue to circumcise their daughters in Nigeria. The results are mixed and further investigation is needed.

A plausible pathway linking maternal education to a daughter's circumcision is through attitudes toward FGM/C. Ideally, education influences attitudes and it is expected that higher educational levels result in support for FGM/C discontinuation [28, 29] and consequently, reduced probability of circumcision among daughters [1, 23]. Studies that test this assumption are scarce. It is unclear the extent to which the associative effect of women's disapproval attitudes and daughters' circumcision varies by maternal educational levels. Furthermore, can an investigation into this explain the mixed results observed in existing findings? Our study addresses these issues by examining data from the 2018 Nigerian Demographic and Health Survey.

## Material and methods

### Data and sampling

We used data from the 2018 Nigeria Demographic and Health Survey (NDHS). It is a cross-sectional and nationally representative survey that collates information on demographic, health and other socioeconomic characteristics of women, men and children in households. Issues on FGM/C were also covered in this survey. Women were asked about their knowledge, attitude and experience of FGM/C as well as that of their daughters. We found the information on FGM/C in the 2018 NDHS data to be substantive in understanding the relationship linking women's education, FGM continuation attitudes and circumcision among daughters.

The 2018 NDHS follows a two-stage sampling design. First, 1,400 clusters were selected from the 2006 Enumeration Area census frame [21]. Second, a fixed number of 30 households were sampled from each of the selected clusters resulting in a total sample size of 42,000 households. Additional details of the sampling design are available in the 2018 NDHS report [21]. Our study used data from the children's and women's recode files of the 2018 NDHS. The children's file contains demographic, health and socioeconomic information on all children that were born in the last five years preceding the survey while the women's file constituted information on all interviewed women from the selected households. Information on circumcision for all daughters born to interviewed women was only available in the women's file. The data for this study was retrieved from the Integrated Public Use Microdata Series—Demographic and Health Surveys (IPUMS-DHS) website at IPUMS-DHS (idhsdata.org). Datasets from the children's and women's files were downloaded separately and later merged for analysis.

The children's file housed information for 33, 924 children below the age of five years. For this study, all male children were excluded and they constituted 50.9% (17,257) of the sample. We further eliminated all female deaths, and this comprised 8.9% (1491) of the remaining sample. An important part of this research is the attitude of mothers toward FGM/C

continuation. Attitudes are susceptible to changes over time or with each successive female birth experience. To ensure that information on attitudes towards FGM/C continuation among mothers during the time of the survey rightly aligns with the circumcision decision of their daughters, we decided to limit our sample to the most recent female birth—youngest daughters born in the last five years preceding the survey. An advantage for doing this was to reduce biases that could lead to reverse causality. In the women's file, information on circumcision was available for a total of 8,471 youngest daughters. After merging this with the children's file, the remaining sample reduced to 5300 youngest daughters who were only born in the last five years preceding the survey. After excluding "missing" and "don't know" responses for the independent variables, our final sample for the analysis constituted 5,039 youngest daughters, born in the last five years preceding the survey and belonging to 1126 clusters.

## Variables

**Outcome variable.**   The main outcome variable for this study is "circumcision among youngest daughters". For each daughter born to a mother, questions on whether they have been circumcised or not were asked. The responses were "No" (0), "Yes" (1) and "Don't know" (7). Our study focused on FGM/C practice among youngest daughters born in the last five years preceding the survey. For this group, the frequency of "don't know" responses was 0. FGM/C among youngest daughters reflects the most recent prevalence of the practice in Nigeria.

**Key independent variables.**   The primary factors for this study are maternal "FGM/C continuation attitudes" and "education". Women were asked if FGM/C should continue. The responses included "Continue", "Depends/Doctor recommends", "Discontinue" and "Don't know". Don't know responses were excluded due to its small sample (<1%). Women were asked about their level of education. Responses included "no education", "primary", "secondary" and "higher".

**Control variables.**   We controlled for some important factors that existing literature have found to be associated with women's FGM continuation attitudes and daughters' circumcision. They were included if information on them were also available in the 2018 NDHS. They are household wealth, place of residence, religion, mother ever circumcised, mother's age, marital status and whether FGM is an important religious practice [1, 22, 23, 33–37]. Richer household wealth enhances women's financial access to resources and information that can consequently influence attitudes and circumcision [1, 35]. Opportunities in urban areas (information, access to social interventions and protection) place women and children at a better advantage against circumcision compared to rural areas [33]. We adjust for the influence of religion since it has been found to be associated with FGM practices and others view it as an important reason for which people engage in the practice [22, 36, 37]. If a mother has ever experienced circumcision, there is a higher likelihood that it will influence her FGM/C attitude and daughters' circumcision [23]. Also, we controlled for marital status and maternal age to account for differences in lifetime FGM/C-related experiences and choices across age and marital groups [1, 37]. All variables and how they were recoded can be found in Table A in S1 Text.

## Statistical analysis

Univariate, bivariate and multivariate analyses were performed to examine the relationships linking mother's FGM continuation attitudes, education and daughter's circumcision as well as the influence of other factors. The test of significant differences was performed using the Pearson's chi-square while percentage distributions were weighted using the sample weighting

factor "PERWEIGHT" in the retrieved IPUMS -DHS data. For the multivariate analysis, we adjusted for clustering using multilevel analysis to reduce the overestimation or underestimation of the regression parameters. We conducted a two-level logistic regression modelling with children at level 1 and clusters at level 2. Since the outcome variable is binary, the level 1 variance was constrained to follow the binomial distribution. For the study's sample, there were 1,126 clusters. To test whether the associative effect of women's FGM/C continuation attitudes on youngest daughter's circumcision differs by educational levels, an interaction term (education * FGM/C continuation) was introduced. We tested for multicollinearity using the Variation Inflation Factor (VIF) and the results showed low collinearity among the explanatory variables (Fig A in S1 Text).

A sequential modelling approach was adopted. The first model was an empty model (Model 0) to ascertain the proportion of the variation in circumcised youngest daughters that is attributable to cluster differences, without accounting for any predictors. Model 1 accounted for the key independent variables and the interaction of these key variables was introduced in Model 2. In Model 3, we controlled for some important factors. Confounders were retained in the final model only if they were significant at 5% alpha level. For each of the models, we computed additional model statistics such as the Akaike Information Criterion (AIC), R—squared, and the intraclass correlation coefficient (ICC). These were useful in determining the best candidate model and the contribution of the models to the overall variation in the circumcision of youngest daughters. The best candidate model has the lowest AIC relative to other models. We also show the proportional change in the variance (PVC) of the random effects to determine the contribution of variables in each model to the remaining cluster variance.

### Ethics statement

The 2018 Nigerian Demographic and Health Survey followed all ethical procedures to ensure that participants' rights were not violated. The survey protocol was reviewed and sanctioned by the National Health Research Ethics Committee of Nigeria (NHREC) as well as the ICF Institutional Review Board [16]. Both written and verbal informed consent were sought from all participants. Consent was also sought from parents/guardian before any information about children were collected. Since the data used for the study was not collected by the authors, access to the data was requested on the DHS Program's website for data use and was approved. Downloaded data was anonymized. The data is freely available upon registration at https://dhsprogram.com/data/available-datasets.cfm or IPUMS-DHS (idhsdata.org).

## Results

### Characteristics of sampled daughters

In Table 1, the weighted percentage distribution of sampled youngest daughters in Nigeria that were born in the last five years preceding the survey by their background characteristics has been presented (univariate). For most daughters, their mothers believed that FGM/C should discontinue (60.5%) and a higher proportion of mothers were not formally educated (44.6%). Most mothers (78.2%) disagreed that FGM/C is an important religious practice and for one-third of children, their mothers were circumcised (32.35). Only an estimated nine percent of mothers were aged 40 years and above. The common religion was Islam (66%) and majority of mothers (95.5%) were in union (either married or living with partner). More than half of the sample resided in rural areas (58.7%) and a substantial number of daughters belonged to poorest households (22.7%).

**Table 1. Weighted percentage distribution of circumcision among youngest daughters by background characteristics.**

| Background characteristics | Univariate | | Bivariate | | $X^2$ |
|---|---|---|---|---|---|
| | | | Youngest daughter circumcised | | |
| | | | No | Yes | |
| | % | n | % [95%CI] | % [95%CI] | |
| FGM continuation attitudes | | | | | 2111.051*** |
| Continue | 31.5 | 1495 | 22.2[20.2,24.3] | 77.8[75.7,80.0] | |
| Discontinue | 60.5 | 3168 | 88.9[87.8,90.0] | 11.1[10.0,12.2] | |
| Depends/Doctor recommends | 8.1 | 376 | 64.7[60.0,69.2] | 35.3[30.8,40.0] | |
| Mother's education | | | | | 435.404*** |
| No education | 44.6 | 2256 | 51.3[49.2,53.3] | 48.7[46.7,50.8] | |
| Primary | 15.3 | 767 | 71.0[67.9,74.2] | 29.0[25.8,32.1] | |
| Secondary | 30.8 | 1553 | 78.0[76.0,80.0] | 22.0[20.0,24.0] | |
| Higher | 9.4 | 463 | 88.0[85.0,90.8] | 12.0[9.2, 15.0] | |
| FGM is an important religious tradition | | | | | 549.368*** |
| Agree | 21.8 | 1120 | 36.6[33.8,39.5] | 63.4[60.5,66.2] | |
| Disagree | 78.2 | 3919 | 74.1[72.8,75.5] | 25.9[24.5,27.2] | |
| Mother circumcised | | | | | 847.925*** |
| No | 67.7 | 3534 | 79.2[77.9,80.6] | 20.8[19.4,22.1] | |
| Yes | 32.3 | 1505 | 38.1[35.8,40.5] | 61.9[59.5,64.2] | |
| Mother's age | | | | | 108.034*** |
| < 25 | 24.0 | 1182 | 54.5[51.2,57.1] | 45.6[42.9,48.4] | |
| 25–29 | 27.4 | 1363 | 69.1[66.7,71.5] | 30.9[28.5,33.3] | |
| 30–34 | 23.6 | 1170 | 71.4[68.9,74.0] | 28.6[26.0,31.1] | |
| 35–39 | 15.7 | 822 | 71.4[68.2,74.5] | 28.6[25.5,31.8] | |
| 40+ | 9.3 | 502 | 63.7[59.5,68.1] | 36.3[31.9,40.5] | |
| Religion | | | | | 564.298*** |
| Christian | 34.4 | 1874 | 87.6[86.1,89.1] | 12.4[10.9,13.9] | |
| Muslim[a] | 65.6 | 3165 | 54.6[52.9,56.3] | 45.4[43.7,47.1] | |
| Marital Status | | | | | 33.724*** |
| Not in Union | 4.5 | 233 | 83.6[78.7,88.3] | 16.4[11.7,21.3] | |
| In Union | 95.5 | 4806 | 65.1[63.8,66.5] | 34.9[33.5,36.2] | |
| Place of Residence | | | | | 161.491*** |
| Urban | 41.3 | 1875 | 76.0[74.1,77.8] | 24.0[22.2,25.9] | |
| Rural | 58.7 | 3164 | 58.9[57.2,60.7] | 41.1[39.3,42.8] | |
| Wealth Status | | | | | 219.217*** |
| Poorest | 22.7 | 1198 | 52.9[50.0,55.8] | 47.1[44.2,50.0] | |
| Poorer | 21.1 | 1075 | 63.0[60.1,65.9] | 37.0[34.1,39.9] | |
| Middle | 20.4 | 1001 | 64.3[61.4,67.2] | 35.7[32.8,38.6] | |
| Richer | 19.3 | 939 | 71.7[68.9,74.5] | 28.3[25.5,31.1] | |
| Richest | 16.6 | 826 | 83.0[80.5,85.5] | 17.0[14.5,19.5] | |
| Overall | **100** | **5039** | **66.0** | **34.0** | |

% Percentages are weighted CI—Confidence interval

*** $p < 0.001$

[a]Muslim category includes a small percentage of Traditional/Other (0.6%)

## Bivariate results

The weighted percentage distribution of sampled daughters by their circumcision status and background characteristics, corresponding 95% confidence intervals as well as Pearson's chi-square are shown in Table 1. Overall, about one-third of all youngest daughters (34%) that were born in the last five years preceding the survey in Nigeria had been circumcised. The Pearson's chi-square test results revealed that the observed differences in circumcision status across all background characteristics were significant. Circumcision among children whose mothers feel FGM/C should continue was high and ranged between 75.7% and 80.0%. There was lower prevalence among youngest daughters whose mothers had attained secondary and higher education at 22.0% and 12.0%, respectively compared to those whose mothers have no education (48.7%).

Among children whose mothers agreed to FGM/C as an important religious practice, there was high prevalence of circumcision and it ranged between 60.5% and 66.2%. Likewise, if a mother had been circumcised, the probability of circumcision for her daughter ranged between 59.5% and 64.2%. The practice was also highest among children whose mothers were aged less than 25 years (45.6%). For children whose mothers were Christians, a higher proportion was not yet circumcised at the time of the survey (87.6%). However, for children whose mothers were Muslims, the proportion of uncircumcised daughters was 54.6%. Among mothers in union, higher prevalence of circumcision was observed among their daughters (34.9%) compared to those not in union (16.4%). Residing in rural areas and belonging to poorest households were characterized by high prevalence of 41.1% and 47.1% compared to other residential and wealth categories, respectively.

## Predictors of FGM/C among youngest daughters

Table 2 shows the odds of being circumcised, their corresponding 95% confidence intervals, random effect and other model statistics. The intraclass correlation coefficient (ICC) in the null model indicates that without accounting for any predictors, 69.1% of the variation in FGM/C among sampled youngest daughters is attributable to cluster differences. In Model 1, we accounted for the key independent variables—FGM/C should continue and education— and both were significant. When these key factors were included in Model 1, the cluster variance reduced by 50.9%. This means that, women's belief on whether FGM/C should continue and their educational background explain half of the variation in the circumcision of youngest daughters that is attributable to cluster differences in Nigeria. Interaction between these two factors was introduced in Model 2 to investigate whether a mother's education is important in influencing the associative effect of her FGM/C continuation attitudes on daughter's circumcision. The interaction was significant. The inclusion of the interaction term increased the remaining cluster variance marginally by 4.3%. Model 3 controlled for the influence of potential confounders. Adding the confounders reduced the remaining cluster variance by 10.4%. When the control variables were introduced, the interaction remained statistically significant at $p < 0.05$. We find Model 3 to be the best candidate model since it had the lowest AIC (3305.9) compared to the other models. The R-squared for this model reveal that an estimated 50% of the variation in circumcision among youngest daughters is explained by the independent variables in the model. The interpretation of the odds ratio is based on Model 3.

We found that the odds of circumcision among youngest daughters were low if the mother who believes FGM/C should discontinue has also attained higher education. There were reduced odds of 0.28[0.08,0.98] of being circumcised among such women compared to those whose mothers approve of FGM/C continuation and have no formal education. Fig 1 shows the predicted probabilities (marginal effects) for the interaction (values for Fig 1 are shown in

**Table 2. Logistic regression results of circumcised daughters by background characteristics.**

| Background characteristics | Model 0 | Model 1 | Model 2 | Model 3 |
|---|---|---|---|---|
| | OR[95%CI] | OR[95%CI] | OR[95%CI] | OR[95%CI] |
| FGM/C continuation attitudes | | | | |
| Continue | | 1.00 | 1.00 | 1.00 |
| Depends/Doctor recommends | | 0.07[0.05,0.11]*** | 0.07[0.05,0.11]*** | 0.09[0.06,0.16]*** |
| Discontinue | | 0.02[0.016,0.029]*** | 0.03[0.02,0.04]*** | 0.05[0.04,0.08]*** |
| Mother's education | | | | |
| No education | | 1.00 | 1.00 | 1.00 |
| Primary | | 0.68[0.48,0.94]* | 0.83[0.48,1.46] | 0.95[0.52,1.73] |
| Secondary | | 0.68[0.51,0.92]* | 1.04[0.64,1.69] | 1.28[0.74,2.21] |
| Higher | | 0.43[0.26,0.70]*** | 1.59[0.53,4.76] | 1.83[0.58,5.79] |
| *FGM/C continuation attitudes: Education* | | | | |
| Continue: No education | | | 1.00 | 1.00 |
| Depends: Primary | | | 1.74[0.57,5.34] | 1.48[0.44,4.98] |
| Discontinue: Primary | | | 0.60[0.30,1.21] | 0.61[0.29,1.28] |
| Depends: Secondary | | | 0.95[0.35,2.58] | 0.99[0.34,2.95] |
| Discontinue: Secondary | | | 0.47[0.26,0.85]* | 0.66[0.36,1.21] |
| Depends: Higher | | | 0.69[0.09,5.15] | 0.98[0.12,8.10] |
| Discontinue: Higher | | | 0.15[0.04,0.50]** | 0.28[0.08,0.98]* |
| **Control variables** | | | | |
| FGM/C is an important religious practice | | | | |
| Agree | | | | 1.00 |
| Disagree | | | | 0.69[0.50,0.93]* |
| Mother circumcised | | | | |
| No | | | | 1.00 |
| Yes | | | | 13.06[9.70,17.58]*** |
| Mother's age | | | | |
| < 25 | | | | 1.00 |
| 25–29 | | | | 0.65[0.48,0.88]** |
| 30–34 | | | | 0.63[0.46,0.88]** |
| 35–39 | | | | 0.72[0.50,1.03] |
| 40+ | | | | 0.82[0.55,1.22] |
| Religion | | | | |
| Christian | | | | 1.00 |
| Muslim[a] | | | | 4.54[3.03,6.79]*** |
| Marital status | | | | |
| Not in Union | | | | 1.00 |
| In Union | | | | 2.50[1.31,4.76]** |
| Place of Residence | | | | |
| Urban | | | | 1.00 |
| Rural | | | | 1.79[1.21,2.65]** |
| Wealth Status | | | | |
| Poorest | | | | 1.00 |
| Poorer | | | | 0.71[0.51,1.00]* |
| Middle | | | | 0.72[0.49,1.06] |
| Richer | | | | 0.58[0.38,0.88]* |
| Richest | | | | 0.53[0.32,0.88]* |
| **Random Effect** | | | | |

*(Continued)*

**Table 2.** (Continued)

| Background characteristics | Model 0 | Model 1 | Model 2 | Model 3 |
|---|---|---|---|---|
| | OR[95%CI] | OR[95%CI] | OR[95%CI] | OR[95%CI] |
| Cluster Variance [SE] | 7.34[0.08] | 3.604[0.06] | 3.76[0.06] | 3.37[0.05] |
| Δ% Cluster variance | - | -50.9 | 4.3 | -10.4 |
| **Model statistics** | | | | |
| Marginal R-squared (%) | 0.00 | 32.5 | 32.7 | 49.9 |
| AIC | 4794.8 | 3715.1 | 3710.0 | 3305.9 |
| ICC (%) | 69.1 | 52.3 | 53.3 | 50.6 |

OR—Odds Ratio CI—Confidence interval

*p<0.05,

**p<0.01,

*** p < 0.001

aMuslim category includes a small percentage of Traditional/Other (0.7%).

Table 3). As observed, daughters of women who believe FGM/C should discontinue have the lowest chances of being circumcised irrespective of mother's education, compared to the other attitude categories. For this group, it can be observed that the probability decreased further by an estimated 40% if the woman has had higher education (3.0%) compared to having no education (5.0%). Among women who believe FGM/C should continue, the probability of their daughter being circumcised worsened for those with secondary (55%) and higher education (64%) compared to those with no formal education (49%). Given this result, we decided to further investigate the characteristics of mothers who believe FGM/C should continue. In our study's sub-sample of only women who believe FGM/C should continue, a higher proportion of those who have secondary and higher education had been circumcised (Table B in S1 Text).

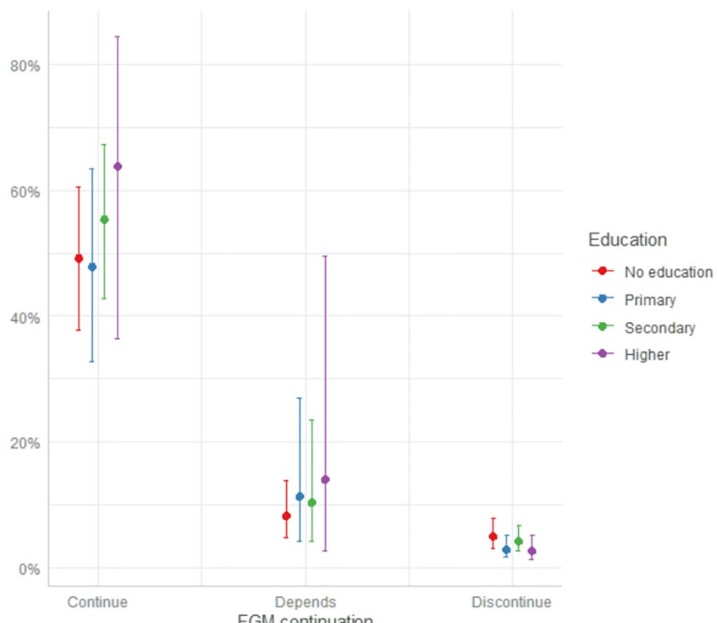

**Fig 1. Predicted probabilities of circumcision among youngest daughters by mother's education and attitude to FGM/C continuation.**

**Table 3. Values for Fig 1.**

|  | Continue | Depends | Discontinue |
|---|---|---|---|
| **No education** | 49.0% [38.0%,61.0%] | 8.0% [0.5%,14.0%] | 5.0% [3.0%,8.0%] |
| **Primary** | 48.0% [33.0%,63.0%] | 11.0% [4.0%,27.0%] | 3.0% [2.0%,5.0%] |
| **Secondary** | 55.0% [43.0%,67.0%] | 10.0% [4.0%,23.0%] | 4.0% [3.0%,7.0%] |
| **Higher** | 64.0% [36.0%,85.0%] | 14.0% [3.0%,50.0%] | 3.0% [1.0%,5.0%] |

The daughters of mothers who disagreed that FGM/C is a religious practice had lower odds of being circumcised, compared to those whose mothers agree that FGM/C is a religious practice (OR = 0.69, 95% CI: 0.50–0.93). If a mother had been circumcised, her youngest daughter had an increased odds of being circumcised as well (OR = 13.06, 95% CI: 9.70–17.58). The associative effect of mother's age on daughter's circumcision was only significant among some specific age groups. Compared to the daughters of women aged less than 25 years, there were statistically significant lower odds of 0.65[0.48,0.88] and 0.63[0.46,0.88] of being circumcised for those whose mothers were within the 25–29 and 30–34 age groups, respectively. Children whose mothers were Muslims (with a small proportion of Traditional and Other religion) had increased odds of 4.54[3.03,6.79] of being circumcised compared to those whose mothers were Christians. There were also higher odds of 2.50 [1.31,4.76] of being circumcised if the mother was in union compared to those whose mothers were not in union. Living in rural areas increased the odds of being circumcised by 79% (OR = 1.79, 95% CI: 1.21–2.65) relative to living in urban areas. Furthermore, living in the richer and richest household was protective as it reduced the odds of the youngest daughter being circumcised by 42% (OR = 0.58, 95%CI: 0.38–0.88) and 47% (OR = 0.53, 95%CI: 0.32–0.88), respectively, compared to living in poorest households. Those that were in the middle and poorer households were not significantly different from those in poorest households. The actual p-values for Model 3 can be found in Table C in S1 Text.

## Discussion

Our study affirms the worrying concerns raised by UNICEF on the rise of FGM/C among young girls in Nigeria [20]. In our study, among the most recent female births in the last five years preceding the survey, a third had already been circumcised. This implies that future prevalence of the practice is likely to have a corresponding increase and reveals inherent relapses that call for more stringent interventions. For most advocacies and social interventions, education has been adopted as a key strategy—either through public education or the formal educational system [38]. Our study investigated the latter and corroborating the findings from other studies [23, 28], we found that individuals who have attained higher education are less likely to circumcise their daughters. The formal education system exposes women to health information and interventions that enhance their knowledge levels and contributes to attitudinal changes [38]. They are also more likely to be empowered to advocate against harmful traditional practices such as FGM/C, thus the evidence in their decision not to circumcise their daughters.

While the formal educational system is key, expanding public education on FGM/C to reach women with no or little education is also crucial. Asekun-Olarinmoye and Amusan [39] conducted an intervention study to examine the impact of health education on attitudes towards FGM/C in a rural Nigerian community and reported that after the health education on FGM was introduced, there were positive changes in FGM attitudes, not only among women but also men. There was a decrease in men who did not want the practice to stop and

an increase in women with no intention to excise future female children. Their study also drew attention to the fact that changing attitudes requires more than just health communication. It also requires skill building and support to enhance individuals' economic capacities. Affirming this is Doucet et al.'s study in Conakry, Guinea [40] who identified that addressing the FGM/C problem involves moving beyond one's "will" to adopt essential empowerment conditions such as the provision of social support and financial independence.

Our study investigated whether the associative effect of FGM continuation attitudes on the circumcision of daughters is partly dependent on maternal education. The results revealed mixed effects. For daughters whose mothers believe FGM/C should discontinue, though the probability of being circumcised was generally low, it further reduced by 40% if the mother had received higher education. This supports the importance of formal education in influencing FGM/C attitudes and practices among daughters. Our findings are, however, contradictory to what De Cao and Giulia [31] found when they explored the impact of the universal primary education programme. De Cao and Giulia [31] reported that the Universal Primary Education programme in Nigeria helped increase the mean years of schooling among women, however, it did not necessarily translate into reduced circumcision among their daughters or drastic changes in their attitude to FGM/C. While their study focused on the contributions of primary education, our findings included those with higher levels of education and show sharp difference. Efforts at the formal education front need to encourage and promote women's education to higher levels.

The results, nevertheless, also revealed that among those who believe FGM/C should continue, the probability of a daughter being circumcised worsened if the woman had attained secondary or higher education (Fig 1). Similar pattern was also observed for those in the "Depends/Doctor recommends" category. The large confidence intervals (Fig 1) show high uncertainty for this result, nonetheless, the estimates reveal nuances that could be explored. Further investigation showed that among the sub-sample of women who believe FGM/C should continue, a higher proportion of those who had attained secondary or higher education had been circumcised (Table B in S1 Text). We therefore consider women's circumcision experience as the underlying (indirect) cause for this observed relationship. Among daughters whose mothers had ever experienced the practice, the odds of being circumcised increased by thirteen folds. This means that despite women's high educational levels, their experience of circumcision can still affect their FGM/C continuation attitudes and consequently, increase the chances of daughter's circumcision. Future studies that explore the role of education on FGM/C need to highlight the boundaries and limitations of women's self-experience of the practice and consider an in-depth look at the socio-cultural processes that come into play to influence women's beliefs and/or choice to circumcise their daughters.

We controlled for the influence of other factors—mother's age, religion, marital status, whether FGM/C is an important religious practice, place of residence and household wealth. Our findings of reducing odds of daughter's circumcision with increasing mother's age contradicts what has been reported in existing literature [37, 41]. In the study's sample, almost half of women aged less than 25 years (45%) had circumcised their daughters and this was high relative to the other age categories. Women of younger ages may be susceptible to lower autonomy levels and strong influence of significant others in favor of FGM/C.

The high prevalence of FGM/C among Muslims has been affirmed in other studies [1, 37]. In this religious group, female circumcision is considered as "Makramah" (means, honourable deed) for women [42] and thus provides some form of identity/belongingness. Therefore, for mothers who believe FGM/C is an important religious practice, daughter's circumcision is high. Towards ending the practice, religious leaders need to be involved in every stage of FGM/C intervention design and implementation.

A perceived social advantage of FGM/C is the matrimonial opportunities it provides and through this, a conduit for FGM/C to persist [6]. Women in union, as found in other studies [1, 43], are more likely to circumcise their children, not only because they have experienced the practice themselves, but also, to enhance their daughter's marriageability, which perhaps, they might have benefitted from. Rural areas and poorest households are nests for FGM/C among girls as affirmed in other studies [1, 33, 35]. These socio-economic conditions are associated with vulnerability, socio-cultural influences and limit to information access which influence women's decision to circumcise or not circumcise their daughters.

FGM/C is a global public health issue and the findings from Nigeria reveal dynamics that could explain evidence found in similar socio-cultural contexts. Our results show that global public health interventions that prioritize female education as a conduit to modifying attitudinal changes regarding FGM/C need to keenly consider circumcision experience of mothers and what that means for them in their social and familial context. The quest to enhance attitudinal and behavioural change towards FGM/C must take into consideration all possible external factors to ensure a holistic implementation of interventions and achievement of desired results.

## Strengths and limitations

Our study has some strengths and limitations. First, it is novel in investigating the pathway linking education, FGM/C continuation attitudes and daughters' circumcision. Our results are also drawn from a nationally representative data which makes our findings generalizable, however, relationships derived are only associations and not causation. Our work was limited to a cross-section of the most recent female births in the last five years preceding the survey. Some girls in the data are still susceptible to being circumcised. We acknowledge the possible influence of social desirability bias since questions on FGM/C—attitudes and experience—were self-reported. Besides, there is the possibility that some FGM/C were not reported due to laws and other factors, thereby leading to underreporting of the overall prevalence. Despite these limitations, the study depicts a nationally representative perspective of the phenomenon.

## Conclusion

The study produces useful insight into how education influences FGM/C which could aid in designing public health educative programs and interventions targeted at addressing FGM/C practices in Nigeria. Education influences FGM/C attitudes, however, women's cutting experience can be a conduit for which FGM/C supportive attitudes persist. Promoting female education should be accompanied by strong political commitment toward enforcing laws on FGM/C practice. Law enforcement agencies, religious leaders and community/opinion leaders need to strengthen their collaborative efforts geared toward the fight against FGM/C in Nigeria. Furthermore, interventions designed to tackle FGM/C could be holistic to consider other socioeconomic and cultural factors such as wealth status, urbanization, religion and beliefs surrounding FGM/C. Programs could also consider community differences in its design to meet the needs of women from diverse communities.

## Supporting information

**S1 Text. Supporting tables and figure.**
(DOCX)

**S2 Text. Inclusivity in global research.**
(DOCX)

## Author Contributions

**Conceptualization:** Josephine Akua Ackah, Bernard Afriyie Owusu, Francis Appiah.

**Data curation:** Josephine Akua Ackah.

**Formal analysis:** Josephine Akua Ackah.

**Methodology:** Josephine Akua Ackah.

**Supervision:** Edward Kwabena Ameyaw, Francis Appiah.

**Writing – original draft:** Josephine Akua Ackah, Patience Ansomah Ayerakwah, Kingsley Boakye, Bernard Afriyie Owusu, Vincent Bio Bediako, Francis Appiah.

**Writing – review & editing:** Patience Ansomah Ayerakwah, Kingsley Boakye, Bernard Afriyie Owusu, Vincent Bio Bediako, Millicent Gyesi, Edward Kwabena Ameyaw, Francis Appiah.

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
