## [Decision Letter · Decision Letter 0]

15 Mar 2022

PGPH-D-21-01063

Circumcising daughters in Nigeria: To what extent does education influence mothers' FGM/C continuation attitudes?

Dear Dr. Ackah,

Thank you for submitting your manuscript to PLOS Global Public Health. After careful consideration, we feel that it has merit but does not fully meet PLOS Global Public Health’s publication criteria as it currently stands. Therefore, we invite you to submit a revised version of the manuscript that addresses the points raised during the review process.

EDITOR: I have reviewed the paper and the reviews submitted from the two other independent reviewers. Please look at the detailed comments and address each in turn. Pay particular attention to the suggestions from reviewer #2 on how to strengthen your discussion.

Please ensure that your decision is justified on PLOS Global Public Health’s publication criteria and not, for example, on novelty or perceived impact.

We look forward to receiving your revised manuscript.

Kind regards,

Colleen M. Davison

Academic Editor

Journal Requirements:

2. Please amend your Financial Disclosure statement. If you did not receive any funding for this study, please simply state: “The authors received no specific funding for this work.”

3. Please update your Competing Interests statement. If you have no competing interests to declare, please state: “The authors have declared that no competing interests exist.”

4. Please note that your Data Availability Statement is currently missing a direct link to access each database. If your manuscript is accepted for publication, you will be asked to provide these details on a very short timeline. We therefore suggest that you provide this information now, though we will not hold up the peer review process if you are unable.

Reviewers' comments:

Reviewer's Responses to Questions

**Comments to the Author**

1. Does this manuscript meet PLOS Global Public Health’s publication criteria? Is the manuscript technically sound, and do the data support the conclusions? The manuscript must describe methodologically and ethically rigorous research with conclusions that are appropriately drawn based on the data presented.

Reviewer #1: Yes

Reviewer #2: Partly

2. Has the statistical analysis been performed appropriately and rigorously?

Reviewer #1: Yes

Reviewer #2: I don't know

3. Have the authors made all data underlying the findings in their manuscript fully available (please refer to the Data Availability Statement at the start of the manuscript PDF file)?

Reviewer #1: Yes

Reviewer #2: Yes

4. Is the manuscript presented in an intelligible fashion and written in standard English?

Reviewer #1: Yes

Reviewer #2: Yes

5. Review Comments to the Author

Reviewer #1: This is a well written manuscript.

The background: should be reduced to about 2 pages. Summarize paragraphs 1 and 2 and present in a paragraph. Do the same in paragraphs 6 and 7. Doing this will bring out study rational clearly.

Methods: Summarize the Statistical analysis section.

Other sections are well described

Little language editing is needed

Reviewer #2: PLOS Global Public Health

ARTICLE: Circumcising daughters in Nigeria: To what extent does education influence mothers' FGM/C continuation attitudes? --Manuscript Draft-- Manuscript Number: PGPH-D-21-01063

Overall, I think this paper is quite important in highlighting the role of education in efforts to eradicate FGM. However, the authors need to go beyond presenting their results and offer a discussion with substance to be published. That is why I consider the recommendation REVISE & RESUBMIT.

INTRODUCTION.

1. Line 66 FGM/C has no health benefit. It is however a clear violation of girls and women’s human rights 67 as it inflicts irreversible and devastating complications such as sexual dysfunction, obstetric 68 complications and psychological trauma among survivors [5],

Authors could reference specific studies done on psychological aspects of FGM/C in Nigeria and elsewhere. See:

Berg, R. C., Denison, E. M. L., & Fretheim, A. (2010). Psychological, social and sexual consequences of female genital mutilation/cutting (FGM/C): a systematic review of quantitative studies. Norwegian Knowledge Centre for the Health Services.

Omigbodun, O., Bella-Awusah, T., Groleau, D., Abdulmalik, J., Emma-Echiegu, N., Adedokun, B., & Omigbodun, A. (2020). Perceptions of the psychological experiences surrounding female genital mutilation/cutting (FGM/C) among the Izzi in Southeast Nigeria. Transcultural psychiatry, 57(1), 212-227.

Omigbodun, O., Bella-Awusah, T., Emma-Echiegu, N., Abdulmalik, J., Omigbodun, A., Doucet, M. H., & Groleau, D. (2021). Escaping Social Rejection, Gaining Total Capital: The Complex Psychological Experience of Female Genital Mutilation/Cutting (FGM/C) Among The Izzi in Southeast Nigeria. BMC Reproductive Health

Mulongo, P., Hollins Martin, C., & McAndrew, S. (2014). The psychological impact of female genital mutilation/cutting (FGM/C) on girls/women’s mental health: a narrative literature review. Journal of Reproductive and Infant Psychology, 32(5), 469-485.

2. Line 84 SDG target 5.3

What does SDG mean?

3. Line 102 Social and behavioural change towards FGM/C has been a major tool to suppressing the practice.

Need to be more specific here. What kind of change are you referring to?

4. Line 125 A plausible pathway linking maternal education to daughter’s circumcision is through attitudes 126 towards FGM/C.

This statement needs to be substantiated in term of the rational as well as supporting references. What about daughter’s education?

5. Line 131 The study produces useful insight on how education influence FGM/C which could 132 aid in designing public health educative programs and interventions targeted at addressing FGM/C 133 practices in Nigeria.

This sentence goes in the conclusion section not the introduction.

METHODS

6. I cannot comment that section as I am a qualitative researcher.

DISCUSSION

1. The interaction between low education level, positive attitude towards FGM and religion as predictors of FGM practice point to the need to discuss the implications of these variables for public health action; in particular, how girls and women can be educated on the deleterious effect of FGM even if they do not access higher level of education.

2. How can religious leader, in particular Muslim leaders, play an education role to help end FGM?

3. How can other types of education on FGM for mothers, fathers and daughter impact the attitude towards FGM?

4. One of my main problem is that the discussion section is underdeveloped when it comes to discussing the results in the light of the literature especially relating to the interaction of the roles played by education, attitudes (personal and cultural) and personal experience of the mother with own FGM. It seems a bit simplistic to focus the discussion only on anti-FGM policy. While policies and anti FGM laws are important they are not enough. Authors data suggest that attitudes, level of education and personal experience with FGM are key. Some qualitative studies have investigated these factors in different countries and should be discussed to enrich and substantiate the discussion section. For example, the mother’s personal experience with FGM was very important in determining attitude of some parents, who refuse to cut their daughter in a context of universal practice of FGM in Guinea. See:

Doucet, M. H., Delamou, A., Manet, H., & Groleau, D. (2020). Beyond will: the empowerment conditions needed to abandon female genital mutilation in Conakry (Guinea), a focused ethnography. BMC Reproductive health, 17(1), 1-15.

5. Authors should suggest the aspects of sociocultural dimension that would need to be studied in relation to these variables. For example, do future studies need to be comparing women with same level of education (high vs low), attitudes towards FGM (for vs against FGM vs ambivalent), self-experience of FGM (positive, negative, traumatic) to see what sociocultural process and context come into play to influence their decision not to cut their daughters. I this context looking at ‘positive deviants’ proved to be heuristic in understanding sociocultural process that come into play and could guide future strategies to eradicate FGM.

6. Line 375. our findings deviate from the conclusions by De Cao and Giulia [23] who focused on the impact of the 376 Universal Primary Education programme on daughter’s circumcision and FGM/C attitudes.

What do you mean here? Please clarify.

7. Also, what are the implications of these results for global public health?

6. PLOS authors have the option to publish the peer review history of their article (what does this mean?). If published, this will include your full peer review and any attached files.

**Do you want your identity to be public for this peer review?** For information about this choice, including consent withdrawal, please see our Privacy Policy.

Reviewer #1: **Yes: **Amelia Ngozi Odo, Ph.D.

Reviewer #2: No

---

## [Decision Letter · Decision Letter 1]

5 Sep 2022

PGPH-D-21-01063R1

Circumcising daughters in Nigeria: To what extent does education influence mothers' FGM/C continuation attitudes?

Dear Dr. Ackah,

Thank you for submitting your manuscript to PLOS Global Public Health. After careful consideration, we feel that it has merit but does not fully meet PLOS Global Public Health’s publication criteria as it currently stands. Therefore, we invite you to submit a revised version of the manuscript that addresses the points raised during the review process.

We look forward to receiving your revised manuscript.

Kind regards,

Colleen M. Davison

Academic Editor

Journal Requirements:

1. Please amend your online Financial Disclosure statement. If you did not receive any funding for this study, please simply state: “The authors received no specific funding for this work.”

Additional Editor Comments (if provided):

The third reviewer (brought on after the first revision) has a few minor edits that the authors should address.

Reviewers' comments:

Reviewer's Responses to Questions

**Comments to the Author**

1. If the authors have adequately addressed your comments raised in a previous round of review and you feel that this manuscript is now acceptable for publication, you may indicate that here to bypass the “Comments to the Author” section, enter your conflict of interest statement in the “Confidential to Editor” section, and submit your "Accept" recommendation.

Reviewer #1: All comments have been addressed

Reviewer #3: (No Response)

2. Does this manuscript meet PLOS Global Public Health’s publication criteria? Is the manuscript technically sound, and do the data support the conclusions? The manuscript must describe methodologically and ethically rigorous research with conclusions that are appropriately drawn based on the data presented.

Reviewer #1: Yes

Reviewer #3: Yes

3. Has the statistical analysis been performed appropriately and rigorously?

Reviewer #1: Yes

Reviewer #3: Yes

4. Have the authors made all data underlying the findings in their manuscript fully available (please refer to the Data Availability Statement at the start of the manuscript PDF file)?

Reviewer #1: Yes

Reviewer #3: Yes

5. Is the manuscript presented in an intelligible fashion and written in standard English?

Reviewer #1: Yes

Reviewer #3: Yes

6. Review Comments to the Author

Reviewer #1: I am satisfied with the revision

Reviewer #3: General comment: Thank you very much for the opportunity to review this manuscript. Overall, the study provides fascinating insights into the link and associative effects between education, FGM/C continuation attitudes, and daughters’ circumcision in Nigeria. The findings have important implications for policy and programmatic interventions for reducing the high prevalence of FGM/C in Nigeria and other West African countries with similar socio-demographic characteristics. I have a few suggestions, comments, and questions that the authors may want to consider in strengthening the manuscript. They are as summarized below:

Methods: Line 161-162: The authors stated that the study considered “youngest daughters who were only born in the last five years preceding the 2018 DHS”. Although this may reflect the most recent prevalence of the practice in Nigeria, in other countries such as The Gambia, the recent DHS reported that while the younger population of children between 0 to 14 years reveals a lower number of 51 percent, 27 percent of children between the age of 0 and 4 years old have been subjected to FGM/C. So, the practice of FGM/C could even happen as soon as the child is born usually at home and not reported by parents for fear of the law, which may not be captured in the most recent DHS.

Results: Line 304-305: There were also higher odds of 2.50 of being circumcised if the mother was in union compared to those whose mothers were not in union. What could be the probable explanation (s) for this finding?

7. PLOS authors have the option to publish the peer review history of their article (what does this mean?). If published, this will include your full peer review and any attached files.

**Do you want your identity to be public for this peer review?** For information about this choice, including consent withdrawal, please see our Privacy Policy.

Reviewer #1: **Yes: **Amelia Ngozi Odo

Reviewer #3: **Yes: **Mat Lowe

---

## [Decision Letter · Decision Letter 2]

25 Oct 2022

Circumcising daughters in Nigeria: To what extent does education influence mothers' FGM/C continuation attitudes?

PGPH-D-21-01063R2

Dear Miss Ackah,

We are pleased to inform you that your manuscript 'Circumcising daughters in Nigeria: To what extent does education influence mothers' FGM/C continuation attitudes?' has been provisionally accepted for publication in PLOS Global Public Health.

Best regards,

Julia Robinson

Staff Editor

Reviewer Comments (if any, and for reference):

Reviewer's Responses to Questions

**Comments to the Author**

1. If the authors have adequately addressed your comments raised in a previous round of review and you feel that this manuscript is now acceptable for publication, you may indicate that here to bypass the “Comments to the Author” section, enter your conflict of interest statement in the “Confidential to Editor” section, and submit your "Accept" recommendation.

Reviewer #3: All comments have been addressed

2. Does this manuscript meet PLOS Global Public Health’s publication criteria? Is the manuscript technically sound, and do the data support the conclusions? The manuscript must describe methodologically and ethically rigorous research with conclusions that are appropriately drawn based on the data presented.

Reviewer #3: Yes

3. Has the statistical analysis been performed appropriately and rigorously?

Reviewer #3: Yes

4. Have the authors made all data underlying the findings in their manuscript fully available (please refer to the Data Availability Statement at the start of the manuscript PDF file)?

Reviewer #3: Yes

5. Is the manuscript presented in an intelligible fashion and written in standard English?

Reviewer #3: Yes

6. Review Comments to the Author

Reviewer #3: (No Response)

7. PLOS authors have the option to publish the peer review history of their article (what does this mean?). If published, this will include your full peer review and any attached files.

**Do you want your identity to be public for this peer review?** For information about this choice, including consent withdrawal, please see our Privacy Policy.

Reviewer #3: **Yes: **Mat Lowe
